# Secretory Leukocyte Protease Inhibitor (SLPI)—A Novel Predictive Biomarker of Acute Kidney Injury after Cardiac Surgery: A Prospective Observational Study

**DOI:** 10.3390/jcm8111931

**Published:** 2019-11-09

**Authors:** Luisa Averdunk, Christina Fitzner, Tatjana Levkovich, David E. Leaf, Michael Sobotta, Jil Vieten, Akinobu Ochi, Gilbert Moeckel, Gernot Marx, Christian Stoppe

**Affiliations:** 1Department of Intensive Care Medicine, RWTH Aachen University Hospital, 52074 Aachen, Germany; luisa.aver@gmail.com (L.A.); cfitzner@ukaachen.de (C.F.); tlevkovich@ukaachen.de (T.L.); msobotta@ukaachen.de (M.S.); jil.vieten@rwth-aachen.de (J.V.); gmarx@ukaachen.de (G.M.); 2Institute of Human Genetics, Medical Faculty, Heinrich Heine University, 40225 Düsseldorf, Germany; 3Division of Renal Medicine, Brigham and Women’s Hospital, Boston, MA 02115, USA; deleaf@bwh.harvard.edu; 4Department of Medicine, Harvard Medical School, Boston, MA 02115, USA; 5Department of Nephropathology, Yale University School of Medicine, New Haven, CT 06510, USA; akiaki5733@med.osaka-cu.ac.jp (A.O.);

**Keywords:** acute kidney injury, cardiovascular surgery, ICU, complications, biomarkers

## Abstract

Acute kidney injury (AKI) is one of the most frequent complications after cardiac surgery and is associated with poor outcomes. Biomarkers of AKI are crucial for the early diagnosis of this condition. Secretory leukocyte protease inhibitor (SLPI) is an alarm anti-protease that has been implicated in the pathogenesis of AKI but has not yet been studied as a diagnostic biomarker of AKI. Using two independent cohorts (development cohort (DC), *n* = 60; validation cohort (VC), *n* = 148), we investigated the performance of SLPI as a diagnostic marker of AKI after cardiac surgery. Serum and urinary levels of SLPI were quantified by ELISA. SLPI was significantly elevated in AKI patients compared with non-AKI patients (6 h, DC: 102.1 vs. 64.9 ng/mL, *p* < 0.001). The area under the receiver operating characteristic curve of serum SLPI 6 h after surgery was 0.87 ((0.76–0.97); DC). The addition of SLPI to standard clinical predictors significantly improved the predictive accuracy of AKI (24 h, VC: odds ratio (OR) = 3.91 (1.44–12.13)). In a subgroup, the increase in serum SLPI was evident before AKI was diagnosed on the basis of serum creatinine or urine output (24 h, VC: OR = 4.89 (1.54–19.92)). In this study, SLPI was identified as a novel candidate biomarker for the early diagnosis of AKI after cardiac surgery.

## 1. Introduction

Acute kidney injury (AKI) is one of the most common complications after major surgery, especially after cardiac surgery [1,2]. Despite substantial improvement in intraoperative management and perioperative care, the incidence of AKI in patients in the intensive care unit (ICU) remains high and ranges between 20% and 67% [3]. AKI necessitates a prolonged ICU stay and is an important prognostic factor of poor mid- to long-term outcomes: it is associated with increased postoperative infections and cardiovascular complications, as well as markedly increased morbidity and mortality rates, even years after surgery [4,5,6,7,8]. Although distinct consensus criteria for the early detection of AKI have been defined by the Kidney Disease Improving Global Outcomes (KDIGO) clinical practice guidelines, AKI continues to be underdiagnosed [9]. To date, the treatment options for AKI are limited, and renal replacement therapy is the standard approach to treating severe cases of AKI. The early identification of patients at risk could enable the timely initiation of preventive measures to reduce the sequelae of AKI [10]. In this context, the currently established and routinely used AKI indicators, such as serum creatinine and urine output, have been repeatedly demonstrated to be insufficient for the early detection of AKI because changes in serum creatinine indicative of altered kidney function are evident only after more than 50% of the baseline renal function has been compromised [11]. Moreover, serum creatinine only serves as a surrogate parameter to estimate the excretory function of the kidney. Thus, serum creatinine does not provide information about the underlying renal pathology and is unable to discriminate between reversible and irreversible injuries [12]. For a better approximation of the extent of injury and the early diagnosis, differential diagnosis, and prognosis of AKI, the international KDIGO board has called for the identification of appropriate AKI markers, analogous to serum troponin or liver enzymes used to identify organ injury [13]. 

Accordingly, studies have investigated several new biomarkers of AKI, and urinary tissue inhibitor of metalloproteinase-2 (TIMP-2) and insulin-like growth factor-binding protein 7 (IGFBP7) ([TIMP-2]·[IGFBP7]) have been deemed promising markers [14]. The assessment of whole-genome mRNA profiles in human kidney biopsies from post-transplant AKI revealed that the most upregulated mRNA (15-fold) was that of secretory leukocyte protease inhibitor (SLPI) [15]. Despite these promising findings, to date, observational studies have not been conducted to evaluate circulating concentrations of SLPI as a biomarker of AKI. Therefore, in a cardiac surgery setting, we tested the predictive value of SLPI as a novel AKI biomarker in two independent prospective observational studies: one development cohort and one validation cohort.

## 2. Materials and Methods

### 2.1. Study Design and Patients

The aim of the study was to investigate the association between serum SLPI levels and the incidence of postoperative AKI after cardiac surgery. The studies, registered at clinicaltrials.gov (NCT 02488876, April 2009), were approved by the institutional review board (Ethics committee, RWTH Aachen University, Aachen, Germany) and performed in adherence to the Declaration of Helsinki. All patients were scheduled for elective cardiac surgery involving aortic cross-clamping, cardioplegic myocardial arrest, and cardiopulmonary bypass. Exclusion criteria were emergency operations, pregnancy, lack of informed consent, an age of less than 18 years, and end-stage renal disease requiring dialysis. 

First, an explorative development study was conducted between September 2015 and March 2016. Second, the results of the development study were used as the basis for a prospective observational validation study, which was conducted from January to June 2017.

Serum creatinine was measured daily. The serum creatinine level on the day before cardiac surgery was used as the reference value. Urine output was quantified hourly by Foley catheter drainage while the patient remained in the intensive care unit.

In both the development and validation studies, serum samples for enzyme-linked immunosorbent assays (ELISA) were drawn one day prior to surgery and immediately (0 h) and 24 h after surgery. In the development study, additional samples were drawn 6 and 12 h after surgery, and, in the validation study, an additional sample was collected 48 h after surgery (Figure 1B).

After blood collection, the samples were centrifuged (3000 rpm for 10 min), and the supernatants were transferred to cryotubes for storage at −80 °C until further analysis. Urine samples were collected preoperatively, immediately postoperatively, and 24 h after surgery and transferred to cryotubes for storage at −80 °C.

### 2.2. Study Endpoints

The primary endpoint of both studies was the development of AKI within 72 h after cardiac surgery. AKI was diagnosed according to the KDIGO clinical practice guidelines by (1) an increase in serum creatinine of at least 0.3 mg/dL or a 50% increase from baseline and/or (2) a decline in urine output to below 0.5 mL/kg/h for at least 6 h [16].

The following patients’ baseline characteristics are known to affect the risk of AKI and were thus determined: age, sex, body mass index (BMI), intake of heart medication, arterial hypertension, pulmonary hypertension, congestive heart disease, reduced left ventricular ejection fraction (LVEF) <35%, chronic kidney disease, chronic obstructive pulmonary disease (COPD), diabetes, and previous cardiac surgery. On the basis of these data, we calculated the Cleveland Clinic Foundation Score—a clinical score used to estimate the risk of developing AKI after cardiac surgery [17]. We recorded the operational characteristics, including the type and duration of surgery, and the postoperative Sequential Organ Failure Score (SOFA) on the first day after surgery (POD1).

### 2.3. Biomarkers

Serum and urine levels of SLPI were measured by ELISA as previously described and according to the manufacturer’s instructions (R&D Systems, Minneapolis, MN, USA) [18]. For the urine samples taken from patients in the development study, we additionally quantified urine neutrophil gelatinase-associated lipocalin (NGAL)—a previously described biomarker of AKI—using a commercially available ELISA kit (R&D Systems, Minneapolis, MN, USA) [19,20]. Before analysis, the serum samples were diluted 1:200 for SLPI ELISA, and urine samples were diluted 1:10 for SLPI ELISA and 1:100 for NGAL ELISA. In a subgroup of 25 patients (the first 25 patients of the validation cohort), we normalized SLPI to the creatinine concentration in the urine. The average coefficient of variation (CV) between duplicates was 9.7% (intra-assay CV) and the average inter-assay cooefficient was 11.9%.

### 2.4. Statistical Methods

Because SLPI serum levels ranged widely in the first study (development study), we performed an additional validation study. The sample size of the validation study was calculated on the basis of the development study. The median of the SLPI levels 24 h after surgery was used as a cut-off. We assumed a power of 90% and set the significance level to 0.05. From the difference in proportions of the “AKI events” in both groups (G1 (≤median SLPI) = 10.7% and G2 (>median SLPI) = 34.5%), we calculated a preferable sample size of 168 patients, assuming a drop-out rate of 25%. The sample size was calculated with PROC POWER, SAS 9.4 (SAS Institute Inc., Cary, NC, USA).

Categorical variables are described by absolute numbers and percentages, and continuous variables are reported as the median and interquartile range (IQR) with the first (Q1) and third (Q3) quartiles.

Differences in baseline characteristics between the two outcome groups were analyzed using univariable logistic regression. Correlations between SLPI and NGAL were calculated using Spearman’s correlation.

The diagnostic accuracy of the biomarkers was calculated by the receiver operating characteristic (ROC) curve and the corresponding area under the curve (AUC). *p*-values were calculated for a hypothesis of AUC > 0.5. Optimal cut-offs were calculated using the Youden index, and 95% confidence intervals, sensitivities, and specificities are reported.

A univariable logistic regression model was used to investigate the performance of SLPI as a predictor of AKI. Given a nonlinear relationship between SLPI and the incidence of AKI, SLPI was considered a binary variable and categorized by the corresponding median to avoid biased estimates [21].

Using multivariable logistic regression models, we adjusted SLPI for the Cleveland Clinic Foundation Score (including sex, congestive heart disease, left ventricular ejection fraction, use of intra-aortic balloon-pump, chronic obstructive pulmonary disease, insulin-requiring diabetes, previous heart surgery, emergency surgery, type of surgery, and preoperative creatinine level) [17]. We applied Firths’ bias reduction implemented in SAS-Macro %fl (SAS Institute Inc., Cary, NC, USA); the odds ratios (OR) with 95% confidence intervals (CI) and *P*-values are reported. To evaluate whether serum SLPI is able to predict AKI before an increase in serum creatinine is evident, we analyzed the time point at which a rise in serum creatinine was detected [22]. Then, in a subgroup analysis, we only considered the patients who received an AKI diagnosis after the time point of SLPI measurement. For example, when analyzing serum SLPI 24 h after surgery, all patients whose serum creatinine had already increased at 24 h after surgery or later were excluded.

In all cases, two-sided testing was used, and *p* <0.05 was considered statistically significant. If not otherwise stated, statistical analyses were performed using SAS Software, version 9.4 (SAS Institute Inc., Cary, NC, USA) and SPSS 25 (IBM SPSS Statistics for Windows, version 21.0. IBM Corp., Armonk, NY, USA).

## 3. Results

### 3.1. Baseline Characteristics and Outcomes of Patients

Of the 70 cardiac surgery patients initially screened for the development study, 60 patients were successfully enrolled. For the validation study, 148 of the 168 screened patients were enrolled (Figure 1A). The incidence of AKI during the first 72 h after cardiac surgery was 25% in the development cohort (DC; 14 of 60 patients) and 15% in the validation cohort (VC; 22 of 148 patients) (Table 1). In both cohorts and for all cases, the diagnostic criterion “increased creatine” was met before oliguria occurred. Oliguria was detected in 21% of AKI cases in the DC and in 23% of AKI cases in the VC (Table 1). In most cases, AKI was diagnosed 48 h after surgery (DC, 50% of cases; VC, 41% of cases) (Table 1). In both cohorts, the overall proportion of AKI patients affected by persistent AKI (>48 h) was approximately 40% (Table 1).

The majority of patients who developed AKI had significantly elevated baseline creatinine levels before surgery (DC: 0.93 mg/dL vs. 1.22 mg/dL, *p* = 0.011; VC: 0.99 mg/dL vs. 1.08 mg/dL, *p* = 0.018) (Table 2). In the development study, AKI was significantly associated with older age (*p* = 0.047), diabetes mellitus (*p* = 0.012), the intake of calcium channel blockers (*p* = 0.037), and an increased Cleveland Clinic Foundation Score (*p* = 0.005). In the VC, AKI was associated with a longer duration of cardiopulmonary bypass (*p* = 0.046, Table 2). No sex-based differences were observed.

### 3.2. AKI Was Associated with Higher Serum SLPI in Cardiac Surgery Patients

After cardiac surgery, serum SLPI significantly increased in both cohorts and peaked at 24 h after surgery (DC and VC: *p* < 0.001). Compared with patients not diagnosed with AKI, those diagnosed with AKI had significantly elevated SLPI serum levels 6, 12, 24, and 48 h after surgery (e.g., 24 h, DC: *p* = 0.001; 24 h, VC: *p* = 0.008; Table 3, Figure 2A,B). Serum SLPI did not differ significantly between transient (<48 h) and persistent (>48 h) AKI cases (Appendix A). Patients with high serum SLPI (higher the median value) 24 h after surgery had a significantly higher incidence of AKI (DC: 10% vs. 38%, *p* = 0.03; VC: 7% vs. 24%, *p* = 0.01; Figure 3). Similar to serum SLPI, urinary SLPI levels were significantly increased 24 h after cardiac surgery (Figure 2C,D). Compared with serum SLPI, urinary levels of SLPI were low overall (approximately 5–10 times lower). After surgery, urinary SLPI levels did not significantly differ between patients with and without AKI (Table 3, Figure 2C,D). When normalized to urinary creatinine, patients with AKI showed significantly higher SLPI levels 24 h after surgery (subgroup of VC, *n* = 25, *p* = 0.01; Figure 2F).

### 3.3. Accuracy of SLPI for Diagnosis of AKI

We assessed the predictive accuracy of SLPI for AKI by the ROC curve and the corresponding AUC. At 24 h after surgery, the area under the ROC curve of serum SLPI was 0.81 (95% CI 0.69–0.92) in the development cohort and 0.69 (95% CI 0.58–0.80) in the validation cohort (Figure 4). The additional earlier time points in the development cohort yielded AUCs of 0.87 (95% CI 0.76–0.97) 6 h after surgery and 0.85 (95% CI 0.74–0.95) 12 h after surgery. When only patients with a Cleveland Clinic Foundation Score ≥3 (“at risk for AKI”) were selected, the diagnostic accuracy did not improve significantly (Appendix A). The AUCs of the absolute increase of SLPI from baseline before surgery (delta from pre-OP) were not superior to absolute SLPI (Appendix A). Compared with urinary NGAL, with an AUC of 0.52 (95% CI 0.31–0.73) 24 h after surgery, serum SLPI was more accurate in diagnosing AKI 24 h after surgery (Figure 4, Appendix A). However, urinary SLPI, even when normalized to urine creatinine, did not demonstrate significant results (AUC = 0.71, 95% CI 0.44–1.0). Table 4 lists the optimal cut-off concentrations calculated using the Youden index and the related sensitivities and specificities.

### 3.4. SLPI as a Predictor of AKI in Univariate and Multivariate Analyses

As calculated in a univariate analysis, patients with higher postoperative serum SLPI (>median) had a significantly higher risk of AKI (e.g., validation cohort, 24 h, OR = 3.89, 95% CI 1.44–12.08, *p* = 0.007; 48 h, OR = 9.24, 95% CI 2.69–48.30, *p* < 0.001; Table 4).

Because the Cleveland Clinic Foundation Score is a clinical score for risk stratification of AKI after open cardiac surgery, this score was included in the multivariable analysis as the reference model (independent variable). The Cleveland Clinic Foundation Score includes the variables sex, congestive heart disease, left ventricular ejection fraction, use of intra-aortic balloon-pump, chronic obstructive pulmonary disease, insulin-requiring diabetes, previous heart surgery, emergency surgery, type of surgery, and preoperative serum creatinine. When adjusted for the Cleveland Clinic Foundation Score, serum SLPI remained a significant predictor of AKI at 6, 12, and 24 h after surgery in the development cohort and 24 and 48 h in the validation cohort (e.g., 6 h, DC: OR = 1.74; 95% CI 1.18–2.84, *p* = 0.004; 24 h, VC: OR = 3.91 95% CI, 1.44–12.13, *p* = 0.007; Table 5).

To determine whether SLPI can be regarded as a predictive biomarker, we performed a subgroup analysis in which we only considered AKI cases that were diagnosed after the respective SLPI measurement. Thus, for SLPI measured at 24 h, we only considered cases of AKI that were diagnosed at 48 or 72 h (*n* = 11 in the development and *n* = 16 in the validation study), and for SLPI measured at 48 h, we only considered cases of AKI that were diagnosed at 72 h after surgery. In addition to the early time points (6 and 12 h after surgery), in these univariable and multivariable analyses, SLPI was significantly predictive of AKI 24 h (multivariable: OR = 4.89; 95% CI, 1.54–19.92; *p* = 0.006) and 48 h (multivariable: OR = 15.24; 95% CI, 1.63–2025.31; *p* = 0.013) after surgery (Table 5). 

## 4. Discussion

SLPI is a 12 kDa (107 amino acids) non-glycosylated single-chain protein that is broadly expressed in myeloid and other epithelial cells [23,24,25]. SLPI functions as a non-redundant alarm anti-protease and is considered important in the defense against proteolytic attack from liberated granulocyte proteases [26]. Apart from its anti-protease activity, SLPI has antibacterial, antiviral, and anti-inflammatory properties and promotes wound healing [27].

Cardiac surgery patients with AKI, even those who achieve complete renal recovery, have a significantly increased risk of death and adverse long-term consequences compared with patients without AKI [8]. Over time, 10%–20% of patients with AKI develop chronic kidney disease [28,29]. The KDIGO clinical practice guideline recommends the early identification of patients at risk and suggests a bundle of preventive measures for the early treatment of kidney injury. Because serum creatinine and urine output show changes in renal function only after the occurrence of significant kidney injury, new biomarkers are needed to earlier identify patients who will later benefit from therapeutic measures [9]. In the 2006 Clinical Path Opportunities List, the Food and Drug Administration declared the identification of new biomarkers as a key area for improving clinical trials and medical therapies [30]. A major limitation in the identification of suitable biomarkers of AKI is the limited availability of human biopsies from kidneys with AKI; therefore, relevant tissue analyses have not yet been conducted in large-scale studies. In post-transplant kidney graft dysfunction, however, the retrieval of kidney biopsies is part of the routine diagnostic panel. The assessment of whole-genome mRNA profiles in eight injured kidney allografts with AKI revealed not only the upregulated expression of established biomarkers such as NGAL but also the significantly enhanced expression of *SLPI* mRNA [15]. The increase in *SLPI* gene expression is correlated with the protein levels of SLPI in the plasma and urine, which indicates a link between elevated SLPI in the urine and blood and the status of the kidney. In fact, immunohistochemical staining and in situ hybridization detected local SLPI protein expression in the kidney tubular epithelial cells, suggesting that the tubule epithelial cells are a source of elevated serum SLPI in patients suffering from post-transplant AKI [31]. Despite the striking baseline-adjusted increase in mRNA expression (15-fold change) that first implicated SLPI as a promising biomarker of AKI, to the best of our knowledge, SLPI has been exclusively tested in the post-transplant AKI setting and has not been examined in AKI after cardiac surgery.

In this prospective observational study, we investigated the possibility of using SLPI to diagnose and predict AKI in patients undergoing cardiac surgery. The incidence of AKI ranged between 15% (in the validation cohort) and 25% (in the development cohort), which is within the normal range of incidence of AKI observed after cardiac surgery [3]. Only a minority of patients met the KDIGO diagnostic criterion of oliguria lasting at least 6 h, which might be attributed to strict counteractive measures, including diuretics and fluid management, undertaken in the ICU setting.

The levels of serum and urinary SLPI considerably increased in the postoperative course and peaked 24 h after the surgical intervention. Compared with healthy blood donors with an average serum SLPI of 49 ng/mL, the serum concentrations of SLPI were approximately twice as high 24 h after cardiac surgery [32]. Contrasting the perioperative values of serum SLPI in the development and validation cohorts, we detected different baseline SLPI levels, whereas the relative changes were comparable. Whether these differences arose from different patient characteristics could not be reliably established.

Patients affected by AKI during the first 72 h after surgery had significantly higher serum SLPI 6, 12, 24, and 48 h after surgery compared with non-AKI patients. SLPI levels exceeding the median 24 h after surgery were associated with a markedly increased risk of AKI. Serum SLPI showed promising accuracy in the diagnosis of AKI, with an AUC of over 0.85 six hours after surgery in the development cohort. However, samples from later time points only yielded moderate results for the AUC and the corresponding sensitivity or specificity of cut-off values. Considering the optimal cut-off values calculated using the Youden index for the different time points, the overall cut-off value of serum SLPI to predict AKI might ranges from around 85 to 90 ng/mL. Compared with patient characteristics that were prognostic for AKI, the addition of SLPI to the risk assessment models significantly improved the prediction of AKI. SLPI was found to detect AKI before a rise in serum creatinine or before decreased urine output became evident. These findings suggest that SLPI is a novel predictive marker of AKI, which may be of particular clinical significance after cardiac surgery but also in a broader intensive care setting associated with AKI.

To date, the functional role of SLPI in the pathogenesis of AKI after cardiac surgery is unknown and requires further examination in experimental studies. In animal models, SLPI has been shown to be an important protective mediator during ischemia–reperfusion injury in the liver and brain, as well as after cardiac transplantation [27,33,34,35]. In a renal ischemia–reperfusion injury mouse model, SLPI was suggested to contribute to tubular cell regeneration via Cyclin-D1 upregulation [36]. Cardiac surgery patients are exposed to a high risk of developing a systemic inflammatory response, which is a substantial factor in the pathogenesis of postoperative AKI [37,38]. After cardiac surgery, SLPI was significantly upregulated and might play a role as a counter-regulatory factor against the detrimental inflammatory response by modulating nuclear factor κ-light-chain-enhancer of activated B cells (NF-κB) and promoting organ repair, such as proximal tubular cell regeneration [35,36,39,40,41]. The sufficient degradation of SLPI in the tubular cells of healthy individuals can be assumed [42,43]. The elevation of serum SLPI levels in acute kidney injury might be a multifactorial and multidirectional resulting from (1) SLPI release from injured kidney tissue, (2) impaired renal elimination, and (3) acute and chronic inflammatory conditions.

One strength of this study is the robust results obtained from two independent and heterogeneous cohorts with significant comorbidities related to AKI, including patients with chronic kidney disease. Compared with some other recently discovered biomarkers, SLPI seems to function accurately as an AKI marker in these unselected study cohorts, which might increase the generalizability of the received findings [44].

Some study limitations exist, and the presented findings need to be interpreted cautiously. Long-term data for the prediction of chronic kidney disease and long-term mortality were not captured in our database and therefore could not be analyzed. Another limitation to our trial is that, for the validation study, only postoperative samples taken 24 and 48 h after surgery were available; thus, the results obtained at 6 and 12 h in the development study could not be compared. However, the results obtained from the development study at 24 h after surgery were successfully confirmed in the validation cohort. Additionally, the appearance that serum SLPI is superior to urinary SLPI as a predictor of AKI should be approached carefully. If a tubular source of SLPI is assumed, then the concentration of SLPI in the urine should be robustly elevated, although we only detected elevated urinary SLPI levels when they were normalized to serum creatinine. As leukocytes are the best-established source of SLPI, elevated SLPI during AKI might, in part, result from a systemic inflammatory response. However, urine concentrations are subject to fluctuations caused by dilution effects (e.g., by diuretics). This might explain why we only observed a significant elevation of urinary SLPI after its normalization to creatinine.

Among the newly identified biomarkers of AKI, urine neutrophil gelatinase-associated lipocalin (NGAL) and [TIMP-2]·[IGFBP7] have provided promising results [14,19,45]. The combined measurement of [TIMP-2]·[IGFBP7] reflects the idea that biomarker panels might better depict the heterogeneous etiology of AKI than single markers. The combination of these identified markers with new biomarkers, such as SLPI, might further improve diagnostic accuracy [46,47,48,49].

## 5. Conclusions

In conclusion, we identified SLPI as a novel biomarker for the early detection of AKI after cardiac surgery. Our findings may lead to future perspectives of biomarker-based risk stratification and may identify patients who would benefit from the early initiation of preventive and treatment strategies.

## Figures and Tables

**Figure 1 jcm-08-01931-f001:**
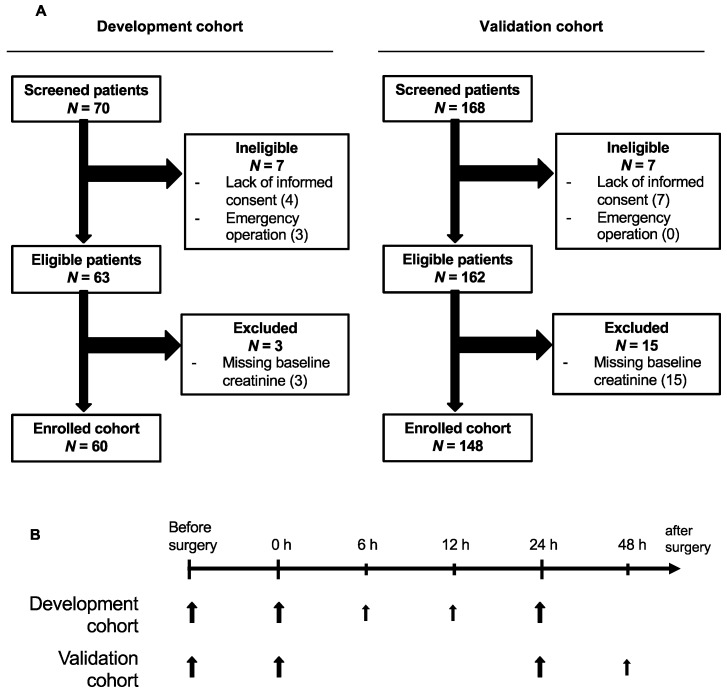
(**A**) Flowcharts of the two independent observational studies investigating SLPI as a biomarker of AKI after cardiac surgery. (**B**) The time points of sample collection for analysis. Larger arrows represent the collection of blood and urine, and smaller arrows represent the collection of blood only. SLPI, secretory leukocyte protease inhibitor; AKI, acute kidney injury.

**Figure 2 jcm-08-01931-f002:**
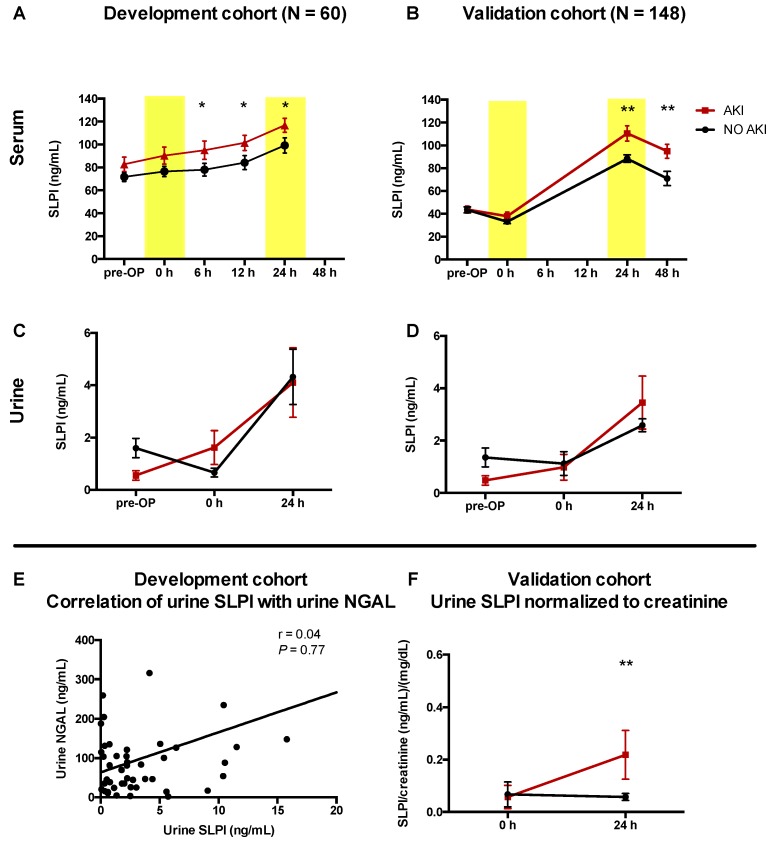
Higher serum SLPI levels were associated with a greater risk of AKI. (**A,B**) Perioperative kinetics of serum SLPI in patients without postoperative AKI compared with patients with AKI. (**C**,**D**) Perioperative kinetics of urinary SLPI. (**E**) Correlation between postoperative urinary SLPI and postoperative urinary NGAL 24 h after cardiac surgery. (**F**) Postoperative kinetics of urinary SLPI normalized to urinary creatinine. AKI, acute kidney injury; NGAL, neutrophil gelatinase-associated lipocalin; Pre-OP, before surgery; SLPI, secretory leukocyte protease inhibitor. Data are means ± SEM; *r*, Spearman’s coefficient. (**A**,**B**) ** p* < 0.05, *** p* < 0.01 versus other groups at the corresponding time point (difference between groups).

**Figure 3 jcm-08-01931-f003:**
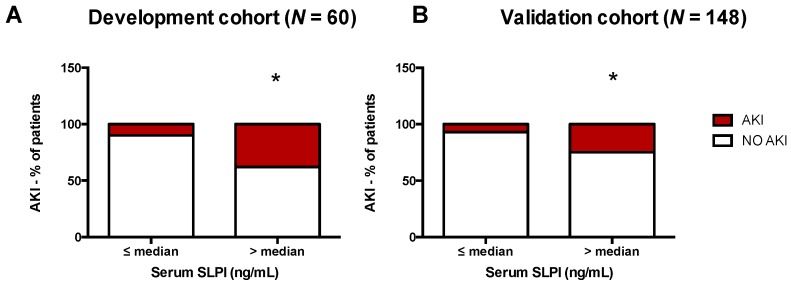
Percentage of patients with AKI within 72 h after cardiac surgery, stratified by median serum SLPI concentration 24 h after surgery (**A**) in the development cohort and (**B**) validation cohort. AKI, acute kidney injury; SLPI, secretory leukocyte protease inhibitor. * *p* < 0.05 analyzed by Fisher’s exact test.

**Figure 4 jcm-08-01931-f004:**
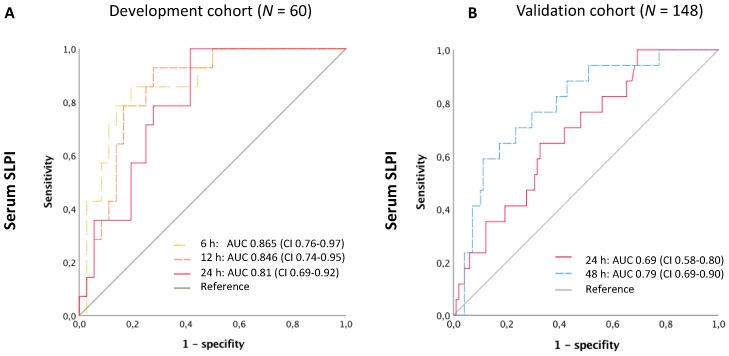
Receiver operating characteristic (ROC) curves of SLPI and NGAL for the diagnosis of AKI at different time points after surgery. (**A**) ROC of serum SLPI in the development study and (**B**) in the validation study. AKI, acute kidney injury; AUC, area under the curve; CI, 95% confidence interval; SLPI, secretory leukocyte protease inhibitor.

**Table 1 jcm-08-01931-t001:** Incidence, diagnostic criteria, and time point of diagnosis of AKI by cohort. Categorical data are presented as the absolute number and percentage. Diagnosis of AKI was based on Kidney Disease Improving Global Outcomes (KDIGO) clinical practice guidelines ((1) an increase in serum creatinine of at least 0.3 mg/dL or an increase of 50% above baseline and/or (2) a decline in urine output to below 0.5 mL/kg/h for at least 6 h) [16]. Most patients diagnosed with AKI were affected by AKI stage 1 and were diagnosed 48 h after surgery. All patients suffering from AKI showed an increase in serum creatinine. Approximately 40% of AKI patients had persistent AKI lasting >48 h. AKI, acute kidney injury.

Acute Kidney Injury within 72 h after Cardiac Surgery		Development Cohort (*n* = 60)		Persistent AKI > 48 h	Validation Cohort (*n* = 148)		Persistent AKI > 48 h
AKI according to KDIGO diagnostic criteria		14	(25%)	6 (43%)	22	(15%)	9 (41%)
	KDIGO Stage 1	8	(57%)		12	(54%)	
	KDIGO Stage 2	5	(36%)		8	(36%)	
	KDIGO Stage 3	1	(7%)		2	(9%)	
Diagnostic criteria met							
	Increased creatinine	14	(100%)		22	(100%)	
	Oliguria (<0.5 mL/kg/h for ≥6 h)	3	(21%)		5	(23%)	
Time point of diagnosis							
	24 h after surgery	3	(21%)	1 (33%)	6	(27%)	2 (33%)
	48 h after surgery	7	(50%)	3 (42%)	9	(41%)	6 (67%)
	72 h after surgery	4	(29%)	2 (50%)	7	(32%)	1 (14%)

**Table 2 jcm-08-01931-t002:** Baseline and operative characteristics by cohort and AKI. Data are expressed as the median (Q1–Q3) or number (percentage). ACE, angiotensin-converting enzyme; AKI, acute kidney injury; BMI, body mass index; CABG, coronary artery bypass graft; COPD, chronic obstructive pulmonary disease; LVEF, left ventricular ejection fraction; POD1, first postoperative day; Q1, Q3, first and third quartile, respectively; and SOFA, Sequential Organ Failure Score. The influence of baseline characteristics on AKI was analyzed by univariable logistic regression. Bold fonts indicate *p*-values < 0.05.

Characteristic		Development Cohort					Validation Cohort				
		No AKI		AKI		*p*-Value	No AKI		AKI		*p*-Value
		(*n* = 46)		(*n* = 14)			(*n* = 126)		(n = 22)		
Demographics											
	Age (years)	67	(59–75)	69	(68–78)	**0.047**	67	(59–75)	69	(68–78)	0.171
	Sex (female)	11	(24)	4	(29)	0.678	33	(26)	6	(27)	0.869
	BMI (kg/m^2^)	27.4	(25.0–29.9)	26.5	(23.8–33.2)	0.767	27.1	(24.8–30.3)	28.5	(22.9–30.4)	0.876
Medication, No (%)											
	Beta blockers	40	(87)	9	(75)	0.292	91	(73)	17	(77)	0.733
	ACE Inhibitors	35	(76)	8	(67)	0.478	69	(55)	12	(55)	0.943
	Sartans	6	(13)	1	(8)	0.840	27	(22)	7	(32)	0.271
	Calcium channel blockers	6	(13)	5	(42)	**0.037**	35	(28)	6	(27)	0.993
	Diuretics	36	(78)	11	(92)	0.456	52	(42)	13	(59)	0.138
	Statins	45	(98)	12	(100)	0.929	106	(85)	19	(86)	0.975
	Acetylsalicylic acid	44	(96)	12	(100)	0.860	103	(82)	18	(82)	0.848
Comorbidities, No (%)											
	Arterial hypertension	28	(61)	11	(85)	0.159	89	(71)	18	(82)	0.362
	Pulmonary hypertension	3	(7)	1	(8)	0.741	6	(5)	2	(9)	0.328
	Congestive heart disease	7	(15)	4	(29)	0.255	16	(13)	0	(0)	0.201
	LVEF < 35%	10	(22)	2	(14)	0.651	6	(5)	2	(9)	0.328
	Chronic kidney disease	3	(7)	2	(14)	0.345	9	(7)	4	(18)	0.090
	COPD	3	(7)	2	(14)	0.345	15	(12)	3	(14)	0.707
	Diabetes, insulin	3	(7)	5	(38)	**0.012**	13	(10)	3	(14)	0.545
	Previous cardiac surgery	3	(7)	0	(0)	0.632	8	(6)	1	(5)	0.970
Serum creatinine at baseline (mg/dL)		0.93	(0.78–1.04)	1.22	(0.83–1.36)	**0.011**	0.99	(0.80–1.10)	1.08	(0.94–1.28)	**0.018**
Type of Surgery											
	Isolated CABG	24	(52)	3	(21)	0.064	78	(62)	11	(50)	0.274
	Isolated valvular surgery	8	(17)	4	(29)	0.344	16	(13)	4	(18)	0.425
	Combined procedure	14	(30)	7	(50)	0.191	30	(24)	7	(32)	0.403
	other						5	(4)	1	(5)	
Risk of AKI											
	Cleveland Clinic Foundation Score	3	(2–3)	4	(3–5)	**0.005**	3	(2–4)	3	(2–4)	0.636
Duration of Surgery											
	Aortic cross clamp	74.5	(57.5–99)	78.5	(47–105)	0.934	73	(55–89)	78	(60–101)	0.232
	Cardiopulmonary bypass	115	(91–144)	118.5	(89.5–148.5)	0.769	109	(87–133)	139	(97–150)	**0.046**
SOFA on POD 1		10	(7.5–12)	9	(7–10)	0.674	8	(6–9)	9	(7–12)	**0.044**

**Table 3 jcm-08-01931-t003:** SLPI measured at different time points. Serum and urinary SLPI concentrations quantified by ELISA and compared between patients with and without AKI. Bold fonts indicate *p*-values <0.05.

**Serum SLPI**										
**SLPI (ng/mL)**	**Development Cohort (*n* = 60)**					**Validation Cohort (*n* = 148)**				
	**No AKI**		**AKI**		***p*-Value**	**No AKI**		**AKI**		***p*-Value**
	**(*n* = 46)**		**(*n* = 14)**			**(*n* = 226)**		**(*n* = 22)**		
Pre-OP	67.3	(57.2–82.1)	87.6	(65.3–98.5)	**0.14**	40.1	(31.6 –48.5)	43.7	(36.6–52.4)	0.280
0 h after surgery	66.3	(52.8–81.15)	102.7	(83.2–128.2)	**0.06**	29.7	(22.4–39.9)	37.9	(25.4–45.3)	0.127
6 h after surgery	64.9	(53.9–84.7)	102.1	(93.2–131.5)	**<0.001**					
12 h after surgery	74.7	(52.0–88.1)	114.5	(95.0–134.5)	**<0.001**					
24 h after surgery	86.1	(69.0–113.5)	117.9	(105.6–145.2)	**0.001**	80.4	(64.7–111.7)	106.6	(83.0–135.3)	**0.008**
48 h after surgery						58.5	(58.5–90.0)	98.8	(76.0–110.4)	**0.000**
**Urinary SLPI**										
**SLPI (ng/mL)**	**Development Cohort (*n* = 60)**					**Validation Cohort (*n* = 148)**				
	**No AKI**		**AKI**		***p*-Value**	**No AKI**		**AKI**		***p*-Value**
	**(*n* = 46)**		**(*n* = 14)**			**(*n* = 226)**		**(*n* = 22)**		
Pre-OP	1.10	(0.40–2.09)	0.40	(0.17–0.96)	0.022	0.51	(0.15–1.53)	0.8	(0.20–1.36)	0.520
0 h after surgery	0.23	(0.07–1.09)	0.58	(0.31–2.02)	0.056	0.13	(0.025–0.35)	0.98	(0.98–1.40)	0.073
24 h after surgery	2.20	(0.74–5.05)	2.38	(0.33–9.23)	0.942	1.15	(0.71–1.92)	1.08	(0.90–1.62)	0.575

Patients with AKI showed significantly elevated serum SLPI after surgery. AKI, acute kidney injury; Pre-OP, before surgery; SLPI, secretory leukocyte protease inhibitor. Data are reported as median (Q1–Q3). *p*-values were analyzed using the Mann–Whitney U test.

**Table 4 jcm-08-01931-t004:** Sensitivity and specificity of SLPI as a biomarker for AKI at optimal cut-off values.

Time Point after Surgery	Optimal Cut-off (ng/mL)	Sensitivity (%)	95% CI	Specificity (%)	95% CI	Likelihood Ratio	Youden Index
Development cohort, Serum SLPI							
6 h	>85.20	64.3	35.1–87.2	68.29	51.9–81.9	2.027	0.32
12 h	>92.72	66.7	34.9–90.1	73.17	57.1–85.8	2.485	0.39
24 h	>87.93	100.0	75.3–100.0	54.55	38.9–69.6	2.200	0.54
Validation cohort, Serum SLPI							
24 h	>101.8	70.0	45.7–88.1	67.6	57.8–76.4	2.162	0.38
48 h	>78.45	77.8	52.4–93.6	71.2	61.4–79.9	2.709	0.49

Optimal cut-off concentrations were calculated with the help of the Youden index; CI, confidence interval; Pre-OP, before surgery; SLPI, secretory leukocyte protease inhibitor.

**Table 5 jcm-08-01931-t005:** SLPI as a predictor of AKI. **(A)** Univariable logistic regression. Serum SLPI was a significant predictor of AKI 12, 24, and 48 h after surgery. Multivariable logistic regression adjusted for Cleveland Clinic Foundation Score (including the variables sex, congestive heart disease, left ventricular ejection fraction, use of intra-aortic balloon-pump, chronic obstructive pulmonary disease, insulin-requiring diabetes, previous heart surgery, emergency surgery, type of surgery, and preoperative serum creatinine). After adjustment for the Cleveland Clinic Foundation Score, serum SLPI remained a significant predictor of AKI. **(B)** Subgroup analysis of cases of AKI that were diagnosed after the respective SLPI measurement. For SLPI measured at 24 h, only the cases of AKI that were diagnosed at 48 or 72 h (*n* = 11 in the DC and *n* = 16 in the VC) were considered. For SLPI measured at 48 h, only the cases of AKI that were diagnosed at 72 h after surgery were considered. SLPI was categorized by the corresponding median; CI, confidence interval; OR, odds ratio; Pre-OP, before surgery; SLPI, secretory leukocyte protease inhibitor; bold fonts indicate *p*-values < 0.05.

**(A) AKI, Time Point not Considered**									
		**Univariable Logistic Regression (Median)**				**Multivariable Logistic Regression (Median)**			
**Time Point after Surgery**	**Median**	**OR**	**95% CI**		***p*-value**	**OR**	**adj. 95% CI**		***p*-value**
Development Cohort									
Pre-OP	71.3	1.37	0.42	4.57	0.601	1.12	0.30	4.16	0.868
0 h after surgery	77.2	2.06	0.63	7.28	0.230	1.69	0.46	6.61	0.431
6 h after surgery	69.6	2.19	0.67	7.82	0.197	**1.74**	1.18	2.84	**0.004**
12 h after surgery	79.9	**3.80**	1.03	17.09	**0.045**	**1.72**	1.15	2.83	**0.008**
24 h after surgery	95	**3.92**	1.10	17.31	**0.035**	**1.76**	1.16	2.98	**0.007**
Validation Cohort									
Pre-OP	41.00	1.46	0.58	3.75	0.417	1.47	0.59	3.76	0.412
0 h after surgery	13.00	1.029	0.41	2.59	0.945	1.01	0.38	2.66	0.19
24 h after surgery	88.3	**3.89**	1.44	12.08	**0.007**	**3.91**	1.44	12.13	**0.007**
48 h after surgery	65.3	**9.24**	2.69	48.30	**<0.001**	**9.45**	2.74	49.55	**<0.001**
**(B) AKI, Time Point Considered**									
		**Univariable Logistic Regression (Median)**				**Multivariable Logistic Regression (Median)**			
**Time Point**	**Median**	**OR**	**95% CI**		***p*-value**	**OR**	**adj. 95% CI**		***p*-value**
Development Cohort									
SLPI measured at 24 h for AKI diagnosed later: 48 or 72 h after surgery (11 of 14 cases of AKI)	95	**4.45**	1.07	25.61	**0.039**	2.48	0.50	15.35	0.268
Validation Cohort									
SLPI measured at 24 h for AKI diagnosed later: 48 or 72 h after surgery (16 of 22 AKI cases)	88.3	**4.94**	1.55	20.15	**0.006**	**4.89**	1.54	19.92	**0.006**
SLPI measured at 48 h for AKI diagnosed later: 72 h after surgery (7 of 22 cases of AKI)	65.3	**15.4**	1.67	2042	**0.011**	**15.24**	1.63	2025.31	**0.013**

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
