# Peer review of "Secretory Leukocyte Protease Inhibitor (SLPI)—A Novel Predictive Biomarker of Acute Kidney Injury after Cardiac Surgery: A Prospective Observational Study"

_jcm, 2019, doi:10.3390/jcm8111931_

Round 1
Reviewer 1 Report
This is an intersting observational study regarding the prognostic value of SLPI for AKI post cardiac surgery. The study has been thoroughly planned, executed and analyzed. I have the following concerns
1. Why urine SLPI was increased pre-op in patients without AKI ? This probalby reflects a selection bias
2. Since NGAL was measured the authors should present a multivariate logistic regression analysis in order to show additional predictive value of SLPI
3. AUC, sensitivity and specificity values are mediocre. The authors should mention this in the discussion section
4. Please describe the term "media" in Table 5
5. The authors should describe a clear cut-off value for SLPI in order their findings to have clinical applicability. Furthermore, the authors should report analytical issues regarding measurement of SLPI (biologic or analytic variation, inter- and intra variability co-efficients)
6. The authors should explain why increase in creatinine was observed before reduction in urine output was seen. Usually in clinical practice clinical picture preceeds laboratory increases in creatinine. Is there a possibility of other than renal causes of increase in creatinine in their study population ?
Reviewer 2 Report
In this prospective observational study of adult cardiac surgery patients, Averdunk et al report the analysis of serine leukocyte protease 1 as it relates to the prediction of AKI. The study has merit in the literature - existing widely used markers of kidney injury are obsolete and non-specific to injury, lagging behind injury -- identifying markers which are capable of high fidelity identification of patients with high risk for AKI would be useful as we bridge the gap towards prophylactic protection and renoprotection.
The study has strengths - 1st time reporting SLP1 (a highly upregulated gene post-ischemic AKI) and the use of both derivation and validation cohorts. However, there are gaps in the existing analysis and to really be novel in the space that has become crowded over the last 10 years with biomarkers predicting AKI, the authors could do a more robust further analysis.
1) The timing of AKI needs to be refined. Recent ADQI panels state clearly we should move toward refining AKI diagnosis as transient (up to but not exceeding 48 hours) from persistent 48h+. If we are predicting actual damage associated AKI, it would be good to separate this out - there are very few patients in the cohort that have AKI post 48 hours. In the prevalence analysis , what happens to the AKI in those patients who develop AKI in the first 24 or 48 hours?
2) Are there longer term outcome data related to the post-operative ICU course?
3) The Cleveland Clinic score is used to adjust for illness severity, but what about determining the biomarker specificity and prediction for AKI in those patients identified by their risk factors at "at - risk"?? this is the model used for "highly successful" biomarkers like troponin - this is has been replicated in the recent publications on renal angina.
4) Would it be possible to combine SLP12 and Creatinine to create a combination biomarker model to separate a post-op creatinine increase from an actual tubular damage associated AKI (ADQI 10)?
5) Do the authors think the absolute SLP1 is more important as a predictor (or better) or the delta from pre-op?
6) There are scattered grammatical issues, but nothing that cannot be fixed easily.
Round 2
Reviewer 1 Report
The authors have addressed adequately my previously raised points
Reviewer 2 Report
Thank you for addressing my concerns.